cognition, behaviour, neuroscience

fMRI, human–computer interaction, intentional stance, social interaction, mentalizing, theory of mind

**Author for correspondence:**
Ahmad M. Abu-Akel
e-mail: ahmad.abuakel@unil.ch

# Re-imaging the intentional stance

Ahmad M. Abu-Akel[1], Ian A. Apperly[2], Stephen J. Wood[2,4,5] and Peter C. Hansen[2,3]

[1]Institute of Psychology, University of Lausanne, Lausanne, Switzerland
[2]School of Psychology, and [3]Centre for Human Brain Health, University of Birmingham, Edgbaston, UK
[4]Orygen, the National Centre of Excellence in Youth Mental Health, Melbourne, Victoria, Australia
[5]Melbourne Neuropsychiatry Centre, University of Melbourne and Melbourne Health, Melbourne, Victoria, Australia

AMA, 0000-0003-3791-8643

The commonly used paradigm to investigate Dennet's 'intentional stance' compares neural activation when participants compete with a human versus a computer. This paradigm confounds whether the opponent is natural or artificial and whether it is intentional or an automaton. This functional magnetic resonance imaging study is, to our knowledge, the first to investigate the intentional stance by orthogonally varying perceptions of the opponents' intentionality (responding actively or passively according to a script) and embodiment (human or a computer). The mere perception of the opponent (whether human or computer) as intentional activated the mentalizing network: the temporoparietal junction (TPJ) bilaterally, right temporal pole, anterior paracingulate cortex (aPCC) and the precuneus. Interacting with humans versus computers induced activations in a more circumscribed right lateralized subnetwork within the mentalizing network, consisting of the TPJ and the aPCC, possibly reflective of the tendency to spontaneously attribute intentionality to humans. The interaction between intentionality (active versus passive) and opponent (human versus computer) recruited the left frontal pole, possibly in response to violations of the default intentional stance towards humans and computers. Employing an orthogonal design is important to adequately capture Dennett's conception of the intentional stance as a mentalizing strategy that can apply equally well to humans and other intentional agents.

## 1. Introduction

To what extent do the cognitive and neural processes involved in playing a competitive game depend on whether one's interactive partner is human? This question bears not only upon theories about the nature of social cognition [1]. It is also of relevance to our everyday lives, in which we interact increasingly with artificially intelligent agents such as robots, computers and avatars, and in which our live interactions with other humans are increasingly mediated through electronic media. A burgeoning literature on social neuroscience has associated such interactions with two types of processes. One is theory of mind (ToM) or mentalizing, which is typically viewed as 'cold cognition' about the beliefs, desires and intentions of one's interactive partner [2]. Mentalizing focuses on the partner's status as a rational, intentional agent, with little regard for the nature of their affective and physical embodiment, and has consistently been associated with activations in the posterior superior temporal sulcus/temporoparietal region (TPJ) and the medial prefrontal cortex (MPFC) (including the anterior paracingulate cortex (aPCC)). A second type of process is mirroring, which is typically viewed as 'flesh-and-blood' simulation of a partner's physical actions and affective reactions [3–7]. In contrast with mentalizing, mirroring focuses on the partner's embodiment, with little regard for the explicit content of their thoughts. While mirroring is commonly associated with activation in

the parietal lobule, the premotor cortex and the inferior frontal gyrus, there is also evidence for mirroring properties in brain regions associated with mentalizing, including the TPJ and the anterior MPFC. In the present study, we investigated the processes involved in playing a competitive game by varying perceptions of the interactive partner's intentionality (they responded freely or according to a script) and their embodiment (they were a human or a computer).

In a very influential paper, titled 'Imaging the intentional stance', Gallagher et al. [8] were the first to employ the human-player versus computer-player contrast in a neuroimaging study. In this study, volunteers were asked to play a version of the 'rock, paper, scissors' (RPS) game against a human opponent or a computer following simple rule-based strategy. In comparing the two conditions (human $_{minus}$ computer), only the aPCC (BA 9/32, bilaterally) was differentially active. Several studies employing various interactive games followed, using a similar script according to which participants were led to believe that they were playing either against a human opponent or a computer [9–14]. For example, stronger activations in the MPFC and the thalamus were observed when playing the Prisoner's Dilemma game against a human than against a computer [10], and that during a matching-pennies game activity within the social brain including the TPJ depended on the extent to which agents were perceived as anthropomorphic or intelligent-looking [14]. Moreover, using a variant of the RPS game, Chaminade et al. [9] reported bilateral activation in the MPFC and TPJ and the right thalamus when contrasting participants playing against a human versus playing a computer generating moves at random. In a further contrast between the human and a robot that participants believed to be endowed with artificial intelligence, only the TPJ was active, leading the authors to conclude that the TPJ was specifically involved in mentalizing about humans. Importantly, in this study, the human-player was presented at all times as an 'intentional agent' with a calculated strategy to win.

However, while these studies drew inspiration from a prominent theoretical account of mentalizing—Dennet's 'intentional stance' theory [15]—they do not accurately capture Dennet's original conception. Dennett [15, p. 17] summarizes the intentional stance as follows 'Here is how it works: first you decide to treat the object whose behavior is to be predicted as a rational agent; then you figure out what beliefs that agent ought to have, given its place in the world and its purpose. Then you figure out what desires it ought to have, on the same considerations, and finally you predict that this rational agent will act to further its goals in the light of its beliefs. A little practical reasoning from the chosen set of beliefs and desires will in most instances yield a decision about what the agent ought to do; that is what you predict the agent will do'. Dennett takes great care to point out that the intentional stance may be adopted towards any object (animal, vegetable or mineral), but that its use naturally depends upon the degree to which that object fulfils the stance's assumption that it is a rational agent. Thus, all there is to being an intentional system is to be a system whose behaviour is well explained by adopting the intentional stance, and which does not necessarily require perceiving the object itself to have mental states (see also [16,17]). In other words, the intentional stance applies just as appropriately to all rational agents, including humans and artificially intelligent robots and computers, and it applies just as inappropriately to humans who lack rationality or free will as it does to pocket calculators. This means that the commonly used paradigm of comparing neural activation when participants compete with a rational human agent and a computer that follows simple rules actually confounds distinct factors: whether the opponent is natural or artificial, and whether the opponent is a rational, intentional agent or an automaton. Existing results in the literature could be owing either to variation in participants' use of the intentional stance for mentalizing, or to variation in their response to interacting with a human versus a computer (via mirroring, or some other process), or to some combination of these. The present study is, to our knowledge, the first in the literature to de-confound these factors in a fully orthogonal design. It should be noted, of course, that orthogonality of the experimental factors does not necessarily imply that the measured functional magnetic resonance imaging (fMRI) blood oxygen level-dependent (BOLD) signal responses will themselves be orthogonal.

Orthogonal variation of the type of competitor and their level of intentionality yields four conditions, the participant plays against one of: an actively competitive human; a non-competitive human who passively follows a predetermined response script; an actively competitive computer endowed with artificial intelligence; and a non-competitive computer that passively follows a predetermined response script. Main effects of the level of intentionality (active versus passive responses) should identify those brain regions involved in deploying the intentional stance in the manner envisaged by Dennett—that is to say, irrespective of whether the target is human or computer. To the degree that mentalizing is well characterized as the adoption of an intentional stance, this main effect should overlap with brain regions commonly associated with mentalizing. Main effects of the type of competitor (human versus computer) should identify brain regions that are distinctively involved in interacting with humans rather than computers, and as described above, the existing literature leads to the prediction that this might involve circumscribed regions within the mentalizing network, namely the TPJ and the MPFC, and perhaps a broader set of regions associated with mirroring. Finally, the interaction between the intentionality and competitor identity factors should identify those brain regions in which the demands of deploying the intentional stance depend on the nature of the competitor. Although the expectation that humans are a special target for mentalizing is a mischaracterization of Dennett's intentional stance theory, there are of course other reasons why this expectation is plausible. Most obviously, humans are surely the most frequent target for mentalizing outside of experimental contexts. This might lead the intentional stance to be the default stance towards humans but not computers. One possible outcome of such a default is disproportionately large effects for the human intentional opponent, while another is that processing may be more effortful when the default must be overcome.

## 2. Methods

### (a) Participants

Twenty-four right-handed, English-proficient healthy adults (five males; 19 females; mean age (s.d.) = 21.21 ± 4.21)) participated in the study. All participants were students from the University of Birmingham. Exclusion criteria included having a history of psychiatric illness, epilepsy, neurological disorders, brain injury as well as current alcohol or substance abuse problems.

**Figure 1.** Each trial began with a countdown 3, 2, 1 in 0.5 s intervals, followed by 'GO' during which the participants made their moves. The 'GO' was present for 1 s followed by a 0.5 s blank screen. The results screen is then displayed for 4 s indicating the moves drawn by both players and the outcome. The winning move is displayed with a yellow star. (Online version in colour.)

## (b) Materials and procedures

In the pre-screening session, English reading proficiency was assessed with the Test of Irregular Word Reading Efficiency (TIWRE) [18] and the Test of Word Reading Efficiency (TOWRE) [19] questionnaires. Handedness was ascertained with the modified Annett Handedness Questionnaire [20]. During the scanning session, participants performed two tasks. The first was a computerized version of the RPS game. The second was Hartwright *et al.*'s [21] British English variant of Saxe & Kanwisher's ToM [22] functional localizer task. At the end of the scanning session, all participants went through a debriefing interview. The study was approved by the University of Birmingham Research Ethics Committee, and written informed consent was obtained from all participants.

## (c) The rock, paper, scissors task

In this task, participants were required to predict the moves of their opponent in order to win. The game has the following simple rules: rock beats scissors, paper beats rock and scissors beat paper. The winner of each round was awarded 1 point. A no-response resulted in an automatic win for the opponent, and identical moves resulted in a draw and no points were awarded. Here, we orthogonally manipulated the intentional stance during the game in such a way that the participants were led to believe that they were playing under four conditions: (i) against an *active* human agent who was a professional RPS player, (ii) a *passive* human agent who followed a predetermined response script, (iii) an *active* intelligent computer program (fictional, called AIRPS) that was capable of analysing the participant's strategy, and (iv) a *passive* computer program that followed a predetermined response script.

Participants were cautioned not to use a stereotyped strategy and to play competitively with the intention of beating their opponent. Feedback was provided during the scan sessions as to how well the participant was scoring at the end of each block of 10 rounds of the game and a summary of the results at the end of each fMRI run. Positive scoring and effort were rewarded with a prize of £10 for the highest performing participant overall at the end of the study. Before each one of the four conditions, participants were provided with on-screen instructions to remind them of what they were required to do and which opponent they would be playing. These instructions were also used to induce a shift in the participant's stance towards their opponent. To reinforce the impression that the participant was truly playing against a 'human' opponent, a 3% fallibility 'no-response' measure was embedded during the human conditions. It is important to note that 'intentional stance' is one among three different stances—the others being design stance and physical stance. There may be degrees of intentional stance, but we are not deliberately investigating these here.

Importantly, unbeknownst to the participants, the game was always played against a computer program generating moves entirely at random. The design ensured that the only difference across the conditions was the particular stance the participant was adopting under the various conditions. Of course, there was always the possibility that participants would not behave in the expected manner under the various conditions. Accordingly, a briefing procedure was used after the scanning session during which participants were asked to recount how they understood and experienced these conditions. This information was gathered to ascertain the intentional stance adopted under the various conditions. Crucially, none of the participants expressed doubt regarding the identity of the four opponents.

The RPS experiment consisted of five fMRI runs, each lasting 440 s per run (approx. 40 min total). Each fMRI run consisted of four blocks, representing the four conditions of interest. The sequence of opponents was chosen from eight predetermined player-sequences (chosen from the 24 possible sequences) such that on each sequence the human and the computer opponents were presented in alternating order. On four of the sequences, the participants' first opponent was a human and on the remaining four a computer. The sequences the participants' played, in each of the five fMRI runs, were selected in a pseudorandom order in the following manner: the first participant played sequences 1 through to 5, the second participants played sequences 6,7,8,1,2, the third participant played sequences 3,4,5,6,7,8 and so forth.

Each block was preceded by a 10 s period during which the instructions were displayed, and followed by a 30 s rest period. During each block, the participant played 10 trials against one of the four possible opponents. Response selections (i.e. rock, paper or scissors) were made using a button box with three active buttons that was placed in the participant's right hand. Figure 1 presents a schematic of stimuli presentation and timing during each trial. All participants went through a practice session of two blocks outside the scanner. The experiment was presented using PRESENTATION (Neurobehavioral Systems, CA, USA), which also recorded the behavioural data (button pressed and reaction time).

**Table 1.** Cluster peaks for the RPS task.

| hemisphere and region | MNI coordinates | | | |
|---|---|---|---|---|
| | X | Y | Z | Z value |
| *active > passive* | | | | |
| L angular gyrus, lateral occipital cortex, temporoparietal junction | −44 | −60 | 32 | 4.94 |
| R angular gyrus, temporoparietal junction, supramarginal gyrus | 56 | −50 | 30 | 4.70 |
| anterior paracingulate cortex | −10 | 44 | 24 | 5.16 |
| L/R precuneous | −8 | −60 | 36 | 3.90 |
| R temporal pole | 28 | 14 | −26 | 4.43 |
| L middle temporal gyrus | −56 | −26 | −14 | 4.39 |
| R middle temporal gyrus | 60 | −20 | −16 | 4.62 |
| *passive > active* | | | | |
| superior parietal lobule | 36 | −52 | 62 | 3.97 |
| *human > computer* | | | | |
| R angular gyrus, lateral occipital cortex, temporoparietal junction | 54 | −62 | 16 | 3.32 |
| anterior paracingulate cortex | 10 | 48 | 30 | 3.43 |
| *computer > human* | | | | |
| L frontal pole | −24 | 44 | −18 | 4.65 |
| R frontal pole | 18 | 46 | −20 | 3.38 |
| *interaction (active computer + passive human) − (passive computer + active human)* | | | | |
| L frontal pole | −34 | 46 | −16 | 4.17 |

## (d) The theory of mind localizer task

This task was used to reliably identify regions within the mentalizing network, which include the TPJ, the paracingulate/MPFC and precuneus and the temporal pole. In this experiment, we used Hartwright *et al.*'s [21] anglicized variant of the Saxe & Kanwisher's task [22] during which participants read 24 short vignettes that were displayed on the screen for 10 s. Half of the stories described the false belief of a character about the current state of affairs (i.e. the false belief (FB) stories), and the other half described a physical event that is non-concurrent with reality such as a photo of a past event (i.e. the false photograph (FP) stories). Each story was followed by a true–false question that was displayed for 4 s, and to which they responded using a response box with two active buttons that was placed in the participant's left hand. The task consisted of four short fMRI runs. In each run, six stories, three FB and three FP, were presented in an alternating order, interleaved with a 12.5 s rest period. All participants went through a practice session of four trials outside the scanner. The experiment was presented using PRESENTATION (Neurobehavioural Systems, CA, USA), which also recorded the behavioural data (response selection and reaction time).

## (e) Functional magnetic resonance imaging data acquisition and analysis

Data were acquired in a single scanning session using a 3 T Philips Achieva scanner. One hundred and seventy-six T2*-weighted standard echo planar imaging (EPI) volumes were obtained in each of the RPS task runs, using a 32 channel head coil. Parameters used to achieve whole-brain coverage are as follows: repetition time (TR) = 2.5 s, echo time (TE) = 35 ms, acquisition matrix = 80 × 80, flip angle = 83°, isotropic voxels 3 × 3 × 3 mm³, 42 slices axial acquisition obtained consecutively in a bottom-up sequence. Using the same parameters, 71 EPI volumes were acquired for each run of the localizer task. AT1-weighted scan was then acquired as a single volume at higher spatial resolution as a three-dimensional (3D) turbo field echo (TFE) image (matrix size 288 × 288, 175 slices, sagittally acquired and reconstructed to 1 × 1 × 1 mm³ isotropic voxels: TE = 3.8 ms, TR = 8.4 ms).

Preprocessing and statistical analyses of the data were performed using the functional magnetic resonance imaging of the brain (FMRIB) software library (FSL v. v. 5.0.6; FMRIB, Oxford, www.fmrib.ox.ac.uk/fsl). For both experiments, initial preprocessing of the functional data consisted of slice timing correction, and motion correction FMRIB's linear image registration tool (MCFLIRT). The BOLD signals were high-pass filtered using a Gaussian weighted filter to remove low-frequency drifts in the bold signal. Spatial smoothing of the BOLD signal was performed using a 5 mm full-width-half-maximum kernel. The functional data were registered to their respective structural images and transformed to a standard template based on the Montreal Neurological Institute (MNI) reference brain, using a 6 degrees of freedom linear transformation (FLIRT).

## (f) Rock, paper, scissors task experiment analysis

Playing against a computer or a human, with either agency or by following a script, provided the four baseline conditions. These four conditions comprised a 2 × 2 ANOVA experimental design with factor 1 being the human versus computer opponent and factor 2 being the element of implied agency from the opponent (active versus passive). Condition regressors were convolved with the canonical haemodynamic response function within a general linear model (GLM) framework. A high-pass filter with a cut-off of 105 s was used. Motion parameters were treated as regressors of no interest in order to account for unwanted motion effects. Session data were aggregated per participant using a second-level fixed effects model. Third-level modelling was used to aggregate the data across participants in a 2 × 2 repeated-measures ANOVA with active versus passive and human versus computer

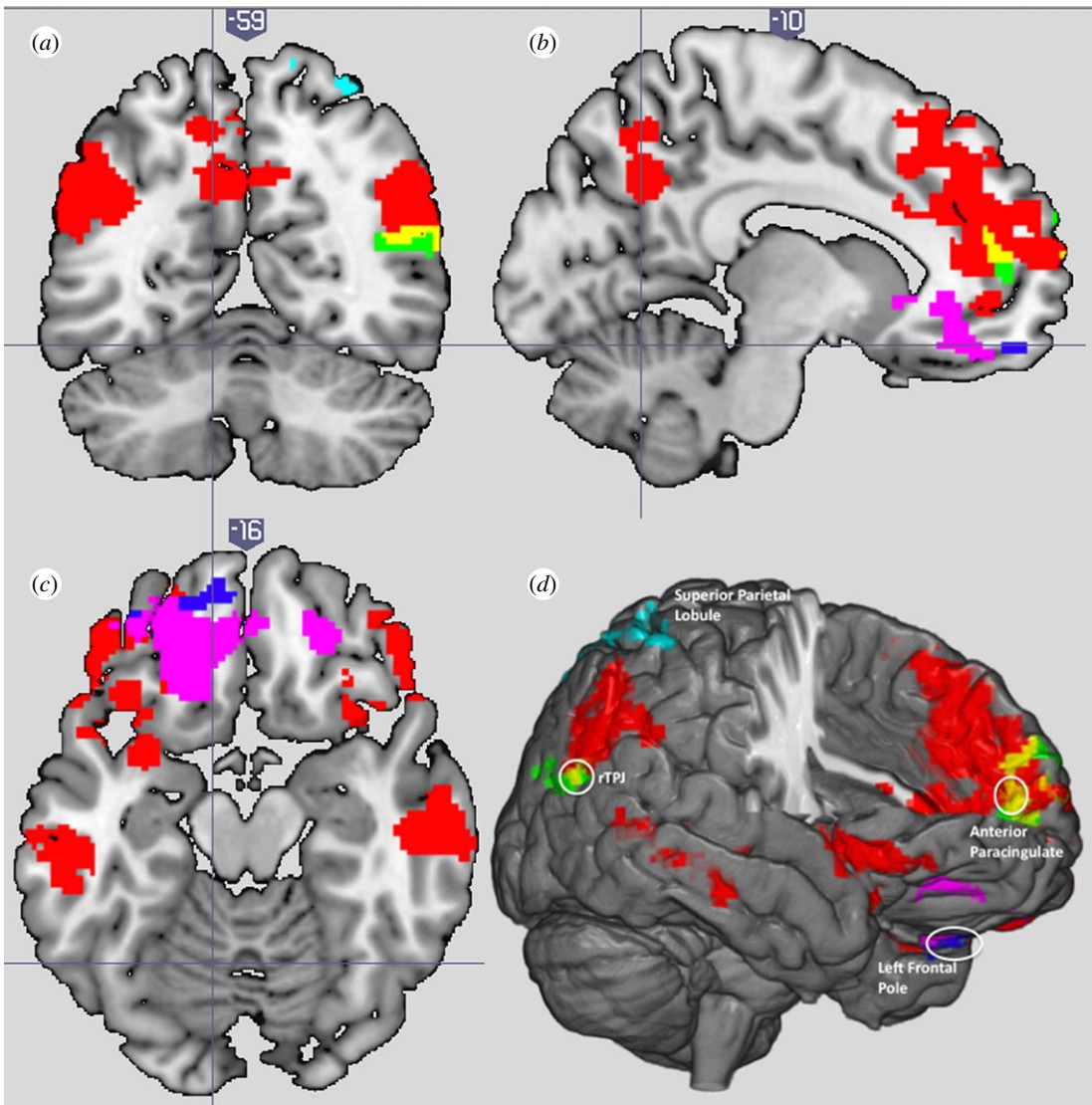

**Figure 2.** Activations of the active $_{minus}$ passive contrast (red), passive $_{minus}$ active contrast (cyan), human $_{minus}$ computer contrast (green), computer $_{minus}$ human (magenta) and the interaction (blue) are presented on (a) coronal ($Y = -59$), (b) sagittal ($X = -10$) and (c) axial ($Z = -16$) planes. Yellow areas in (a,b,d) reflect overlapping areas between the active $_{minus}$ passive and the human $_{minus}$ computer contrasts. The overlap analysis between the active $_{minus}$ passive and human $_{minus}$ computer revealed shared activation in the paracingulate [$-2$, 54, 8] and the rTPJ [50, $-60$, 18] ($Z > 2.3$, $p$corr < 0.05). (d) An annotated 3D summary image. Images are displayed in neurological convention, where left is represented on the left side of the image. (Online version in colour.)

as within-subjects factors, employing a mixed-effects analysis with cluster-based thresholding at $Z > 2.3$, $p$corr < 0.05.

## (g) Localizer task experiment analysis

The localizer task was modelled as per Hartwright *et al.* [21]. The FB and the FP conditions were convolved with a gamma-derived canonical haemodynamic response function within a GLM. A high-pass filter with a cut-off of 21 s was used. Second- and third-level modelling were used to aggregate the data across sessions and participants for the contrast of interest FB > FP. Individual's participant session data were aggregated using a fixed effects model at second level, and the group data were aggregated at third level using a mixed-effects analysis with cluster-based thresholding at $Z > 3.6$, $p$corr < 0.05.

## (h) Overlap analysis

Overlap analysis between the thresholded data ($Z > 2.3$, $p$corr < 0.05) for the human > computer and the active > passive contrasts was conducted to identify shared activations across the two thresholded contrasts. We also conducted an overlap analysis between these two contrasts and the FB > FP contrast of the ToM localizer task. The analysis was conducted with FSL's *easythresh* function [23].

## 3. Results

### (a) Behavioural results

Performance on the RPS task was examined using a 2 (human versus computer) × 2 (active versus passive) repeated measures. The analysis revealed non-significant main effects (active versus passive; $F_{1,23} = 1.87$, $p = 0.18$; human versus computer; $F_{1,23} = 2.19$, $p = 0.15$) or interaction $F_{1,23} = 0.17$, $p = 0.68$), suggesting that participants won equally likely across all conditions.

With respect to the FB task, a pair-*t*-test revealed that there were no differences between the FB and FP conditions in either proportion correct responses ($t_{22} = 1.69$, $p = 0.11$) or reaction times ($t_{22} = 1.25$, $p = 0.23$).

### (b) Whole-brain analysis

#### (i) Rock, paper, scissors task

A 2 × 2 repeated-measures ANOVA of the RPS task identified main effects of the game partner (computer versus human) and intentionality (active versus passive), as well as an

**Table 2.** Cluster peaks for the ToM localizer task.

| hemisphere and region | MNI coordinates | | | |
|---|---|---|---|---|
| | X | Y | Z | Z value |
| *false belief > false photograph* | | | | |
| L angular gyrus, lateral occipital cortex, supramarginal gyrus, temporoparietal junction | −56 | −62 | 28 | 6.18 |
| R angular gyrus, lateral occipital cortex, temporoparietal junction | 56 | −64 | 30 | 5.22 |
| L/R paracingulate cortex, frontal pole | 0 | 58 | 10 | 5.87 |
| L/R precuneus | 0 | −60 | 30 | 6.85 |
| L/R cingulate cortex | 0 | −16 | 34 | 5.59 |
| R medial frontal gyrus | 44 | 12 | 50 | 4.54 |
| R frontal orbital | 50 | 30 | −16 | 4.72 |
| L inferior/middle temporal gyrus | −50 | 0 | −40 | 5.81 |
| R inferior/middle temporal gyrus | 50 | 0 | −38 | 6.43 |
| L cerebellum crus II | −28 | −84 | −36 | 6.25 |
| R cerebellum crus II | 30 | −86 | −36 | 5.09 |
| L cerbellum IX, vermis VIIIb | −6 | −62 | −44 | 5.04 |
| L amygdala | −18 | −4 | −20 | 4.08 |

interaction between the two factors. Playing an active rather than a passive opponent largely recruited a network of regions associated with mentalizing, which included the TPJ bilaterally, right temporal pole, aPCC and precuneus. In addition, the middle temporal gyri were activated. The reverse contrast of passive $_{minus}$ active revealed activation only in the superior parietal lobule. Playing a human rather than a computer, activations were observed in the right TPJ and the aPCC only. Here, the reverse contrast of computer $_{minus}$ human revealed bilateral activations in the frontal pole. Intriguingly, the interaction between the implied agency (i.e. whether the opponent is active or passive) and the game partner (i.e. whether the opponent is computer or human) elicited activation in the left frontal pole only, specifically in the (active computer × passive human) $_{minus}$ (passive computer × active human) contrast (table 1 and figure 2).

### (ii) Theory of mind localizer task

The mixed-effect analysis of the FB $_{minus}$ FP contrast revealed activations in core regions within the prototypical mentalizing network which included both the left and right TPJ, the precuneus as well as the MPFC (table 2). These results are consistent with previous studies using this task [21,22].

### (c) Overlap analysis

The overlap analysis between the active $_{minus}$ passive and human $_{minus}$ computer revealed shared activation in the paracingulate [−2, 54, 8] and the rTPJ [50, −60, 18] (figure 2). The human $_{minus}$ computer overlapped with the FB $_{minus}$ FP contrast at the right medial frontal gyrus [10, 48, 30], in the vicinity of the paracingulate cortex. Finally, the activations maps in the active $_{minus}$ passive and the FB $_{minus}$ FP contrasts overlapped in the TPJ bilaterally [−44, −60, 32; 58, −52, 28], and the paracingulate cortex [−4, 50, 20] (figure 3).

## 4. Discussion

In the present study, we examined the brain regions recruited when people play an interactive game against an opponent that they took to be either a human or a computer, and either freely intentional or passively following a script. As such, this is, to our knowledge, the first study in the literature to de-confound these factors in a fully orthogonal design. A key finding of our study is that a network of regions involved in mentalizing was activated whenever participants believed their opponent to be an intentional agent, irrespective of whether they believed them to be a human or a computer. As presented in table 2 and visualized in figure 2, the main effect of intentionality bilaterally activated the TPJ, the precuneus, the aPCC and the right temporal pole. Converging evidence that these are indeed brain regions consistently implicated in mentalizing came from the substantial overlap between these brain regions and those observed during the ToM localizer task (figure 3). It has been suggested that the correspondence of these brain regions with the mentalizing network may reflect that the participants tried to mentalize an opponent's intention, tactics and emotion during the game [14]. These results are clearly consistent with Dennett's [15] notion of an intentional stance that applies equally well to human and non-human intentional agents.

The second main contrast, human $_{minus}$ computer, revealed brain activity that was confined to the rTPJ and aPCC. In principle, one interpretation of results from this contrast would have been that it revealed effects owing to simulative mirroring [24,25] of the human opponent that is not applied to the computer opponent because it lacks the participants' embodiment. However, this interpretation fits less well with the absence of any observation in the present study of activity in the premotor cortex or inferior frontal gyrus, which might have been expected if participants were simulating the actions of their human competitors [25]. The absence of these and other 'mirroring' effects does not count against an important

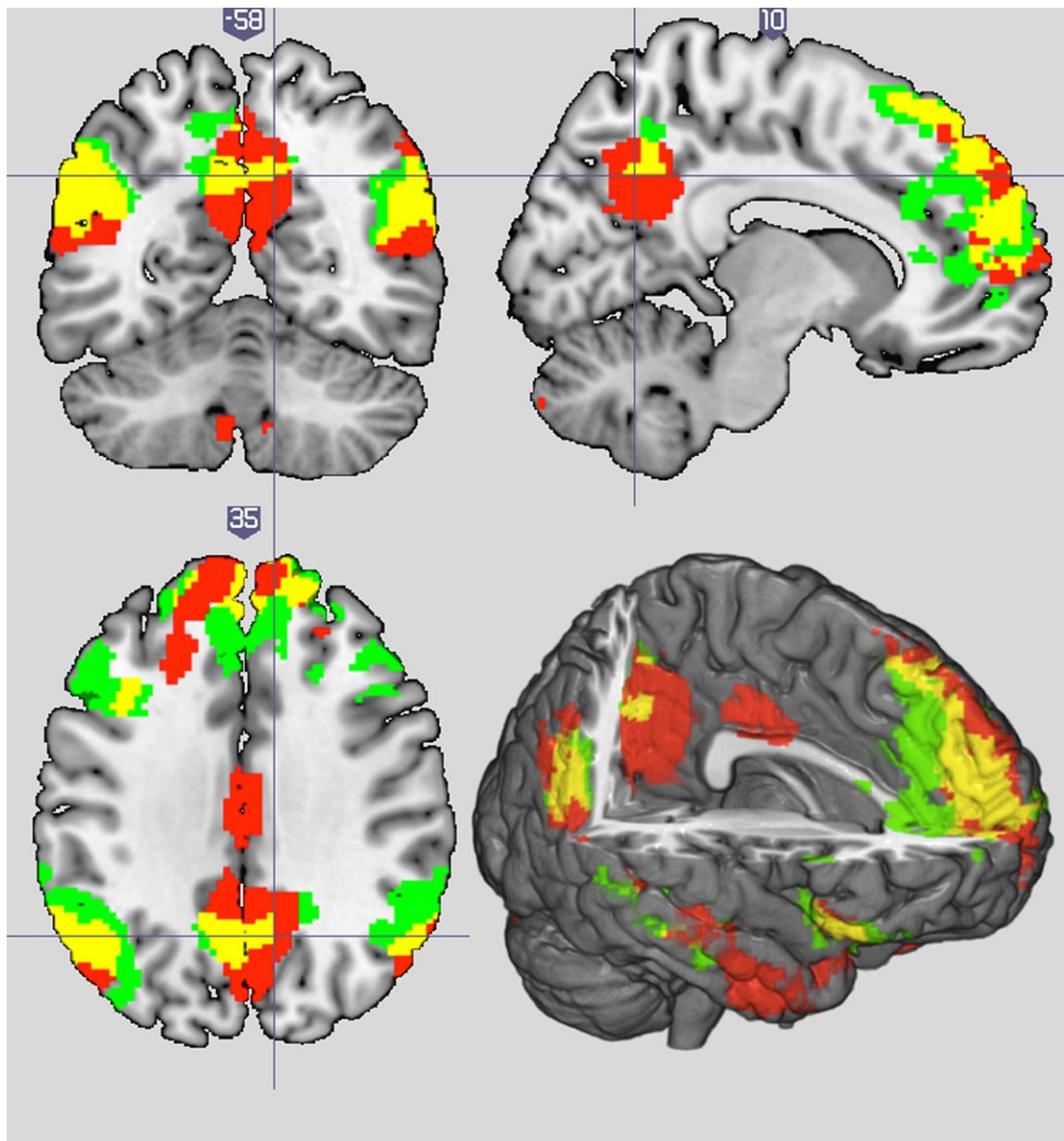

**Figure 3.** Overlaps between the FB ₘᵢₙᵤₛ FP (ToM localizer task; red) and the active ₘᵢₙᵤₛ passive contrast (RPS task, green), presented on coronal ($Y = -58$), sagittal ($X = 10$) and axial ($Z = 35$) planes. Yellow reflects overlapping areas between the active ₘᵢₙᵤₛ passive and the FB ₘᵢₙᵤₛ FP contrasts. Only the left TPJ [−44, −60, 32], right TPJ [58, −52, 28] and paracingulate cortex [−4, 50, 20] survived thresholding ($Z > 2.3$, $p$corr $< 0.05$). Images are displayed in neurological convention, where left is represented on the left side of the image. (Online version in colour.)

role for mirroring in social cognition more generally, and may make sense in the current study, given that participants were never able to observe their competitor or their actions. This is consistent with the results of a large meta-analysis (of over 200 fMRI studies) showing that the mirror network activated in the presence of observable biological motion and the mentalizing network activated when individuals inferred the intentions of others based on abstract information and in the absence of any perceivable biological motion [26]. As such, this leads us to suggest that the observed activity in rTPJ and aPCC for the human ₘᵢₙᵤₛ computer contrast reflects spontaneous mentalizing, rather than mirroring. These two regions have been consistently activated during spontaneous mentalizing in other studies [27,28] and have previously been shown to respond preferentially to action/stimuli that are deemed of human (versus computer) origin [29,30]. Consistent with claims that people have a basic tendency to differentiate humans and computers along the lines of intentionality [31], we suggest that the mere presence of the human competitor in the present study was sufficient to cue participants to

think about their mental states, even though the passive human competitor had no opportunity to deploy these strategically in the game.

The third contrast of principal interest was the interaction between intentionality (active versus passive) and agent (human versus computer). Recall that one natural prediction from the hypothesis that humans are a default target for mentalizing is that activity in brain regions associated with mentalizing will be disproportionately high for the active human condition. In fact, the only brain region identified with the interaction analysis was the left frontal pole (specifically at the base of the frontal pole ~BA 11). This region has occasionally been reported in studies of mentalizing [32,33], but it was not identified either by the main effect of intentionality or by the ToM localizer in the present study. However, the left frontal pole has frequently been implicated in inhibitory control and the suppression of distractions that interfere with the execution of goal-directed actions [34], as well as in evaluative reasoning, in which salient but logically incorrect alternatives must be ignored [35]. We propose that this

activity can be understood in terms of the hypothesis that humans are a default target for mentalizing, not because the left frontal pole is involved in mentalizing *per se*, but because it is recruited for overcoming this processing default. As stated above, people have a basic inclination to differentiate humans and computers along the lines of intentionality [31] and to respond preferentially to stimuli and actions that are generated (or believed to be so) by humans [30]. Thus, if the default is to employ an intentional stance towards a human and an instrumental or a physical stance towards a computer, then the interaction observed in the left frontal pole may reflect the need to deploy a different stance to the one normally adopted to the interacting partner. This interpretation may also be extended to account for the deployment of the bilateral frontal pole in the computer $_{minus}$ human contrast by making the plausible assumption that participants have a general default to employ an intentional stance when playing a competitive game such as RPS which is compatible with their default for a human competitor but not their default for a computer competitor. One can also conjecture that the observed effect for the frontal pole is owing to executive functions involved in the RPS game itself, for example, in the attempts of inhibiting a natural tendency to repeat the same move as the opponent, which would be consistent with the role of the frontal pole in manipulating and maintaining information from self-generated behaviour [36]. Such processes might be specifically strongly activated where participants might have been particularly competitive. In all, the involvement of such control regions is consistent with theoretical accounts, suggesting that the attribution of intentionality and agenthood is a flexible process, and that such flexibility in the attribution of intentionality (whether to active or passive, human or computer agents) can be manipulated volitionally and even strategically [37], but that such strategic deployment works either with or against the default stance for a particular target or activity.

The fact that playing a human competitor only activated a subset of the regions of the 'mentalizing network' activated when playing an intentional competitor (whether human or computer) may be informative about the different function of these brain regions for mentalizing. The recruitment of additional regions when playing an intentional competitor may reflect the difference between 'mere mentalizing', that does not require the integration of mental states in an online activity, and the *use* or deployment of mentalizing, that systematically draws on memory for task-relevant information (e.g. what the opponent did last time; or how the identity of the agent might determine her strategy) or resolves task-relevant conflict (between the opponent's intentions and the participant's own). These additional regions have been variably activated in a variety of ToM tasks [38], and appear to play a general role within the broader mentalizing network [38]. For example, while there is some causal evidence that the left TPJ is as important as the rTPJ for processing mental states [39], it appears to have a more general role in processing perspective difference for both mental and non-mental states [40]. The precuneus has been implicated in processing autobiographical memory and visuospatial attention [41], and the temporal pole is involved in face recognition and schematic knowledge of social memory [42]. In addition, a closer look at activations of the rTPJ across both contrasts reveals that participants recruited both the angular and the supramarginal gyri in the active $_{minus}$ passive condition, and only the angular gyrus in the human $_{minus}$ computer condition. This is confirmed in the overlap analysis where the shared activation is in the angular gyrus. In this regard, it has been proposed that the angular gyrus is selectively involved in social cognition ('mere mentalizing', in the present study), whereas the supramarginal gyrus is more involved in attention reorienting [43], which is likely to be essential for *use* of mentalizing for any practical purpose (see also [12]).

In conclusion, our results indicate that activation of the 'mentalizing network' might be specific to mentalizing, but it is not specific to mentalizing about humans and can be activated by the mere belief that the target (human or not) is a thinking entity. Interacting with humans versus computers, however, induces activations in a more circumscribed right lateralized subnetwork within the mentalizing network, consisting of the rTPJ and the aPCC, and which might be reflective of people's spontaneous tendency to attribute intentionality to humans. Interestingly, frontal control regions appear differentially active in response to violation of the default stance adopted to the target, and the degree to which we readily attribute human-like abilities to the target. Together, these findings expand on earlier results from research investigating human–computer interactions in various social games [10,11,13,14,44]. They emphasize the importance of employing an orthogonal design to adequately capture Dennett's conception of the intentional stance and, consistent with Dennett's view, suggest that the same neural mechanisms are recruited for mentalizing irrespective of the nature of the target.

**Ethics.** The study was approved by the University of Birmingham Research Ethics Committee, and written informed consent was obtained from all participants.

**Data accessibility.** The data that support the findings of this study are available from the Dryad Digital Repository: https://dx.doi.org/10.5061/dryad.j3tx95x9f [45].

**Authors' contributions.** A.M.A.-A. contributed to the design of the study, collected and analysed the data and wrote the manuscript. I.A.A. and P.C.H. designed the study. P.C.H. contributed to the conception of analytical approach. All authors discussed the results and commented on the manuscript.

**Competing interests.** We declare we have no competing interests.

**Funding.** This research received support from the Economic and Social Science Research Council (ESRC), grant no. ES/J012238/1.

**Acknowledgements.** We thank Hannah Widdman for helping prototype an initial version of the RPS task.

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
