## [Reviewer comments · Proceedings of the Royal Society B: Biological Sciences]

Review History

RSPB-2019-1553.R0 (Original submission)

Review form: Reviewer 1

Recommendation

Major revision is needed (please make suggestions in comments)

Scientific importance: Is the manuscript an original and important contribution to its field?

Good

General interest: Is the paper of sufficient general interest?

Good

Quality of the paper: Is the overall quality of the paper suitable?

Acceptable

Is the length of the paper justified?

Yes

Should the paper be seen by a specialist statistical reviewer?

Yes

Do you have any concerns about statistical analyses in this paper? If so, please specify them explicitly in your report.

No

It is a condition of publication that authors make their supporting data, code and materials available - either as supplementary material or hosted in an external repository. Please rate, if applicable, the supporting data on the following criteria.

Is it accessible?

N/A

Is it clear?

Yes

Is it adequate?

Yes

Do you have any ethical concerns with this paper?

No

Comments to the Author

The scope of this paper is interesting. However, if authors would like to separate two compound factors involved in "intentional stance", they should define these factors more strictly.

To my understanding, we must ideally prepared orthogonal independent variables in fMRI analysis (if not, these variables might eat same brain signals and the result might be distorted). Although, authors claimed they prepare two orthogonal independent variables, they should guarantee these two different variables are rigid and independent ones from experimental data.

Recent studies have suggested that strategic aspects of participant's behaviors also affected brain activities in interactive games. If authors purely want to argue about "the stance" of participants, they should conduct similar behavioral analysis in following previous articles and exclude brain signals caused by the behavioral difference among conditions. Even if authors claim that "stance" also contain behavioral aspects, they should do computational behavioral analysis in line with modern neuroscience.

[ref]

Hampton, A. N., Bossaerts, P., & O'Doherty, J. P. (2008). Neural correlates of mentalizing-related computations during strategic interactions in humans. *Proceedings of the National Academy of Sciences*, 105(18), 6741-6746.

Lee, D., McGreevy, B. P., & Barraclough, D. J. (2005). Learning and decision making in monkeys during a rock-paper-scissors game. *Cognitive Brain Research*, 25(2), 416-430.

Takahashi, H., Izuma, K., Matsumoto, M., Matsumoto, K., & Omori, T. (2015). The anterior insula tracks behavioral entropy during an interpersonal competitive game. *PLoS one*, 10(6), e0123329.

Furthermore, there are few information about participants. The attitude toward "human" and "computer" were supposed to be varied among people. The various factors (e.g. education, personalities, culture...) shape our attitude and there are huge diversity in it. Hence if authors merely merge data regardless of individual differences, it is not so strange that the identities about opponent did not predict brain signal well (because S/N of data is too bad). Please describe precise information about participant's identities (e.g. education, Expertise) and authors should show the attitudes toward "human" and "computer" was not so varied in this experiment.

Did participants ask some questionnaires about the impression of opponents? Previous studies

have suggested the impression of other agent lineally predicted the activation of mentalizing network. The resolutions of words "human" and "computer" are too low (e.g. the image of computer might to be different between participants with the knowledge about AlfaGO and people who did not know it). Authors should quantify participant's stance toward different two opponents.

{ref]

Takahashi, H., Terada, K., Morita, T., Suzuki, S., Haji, T., Kozima, H., ... & Naito, E. (2014). Different impressions of other agents obtained through social interaction uniquely modulate dorsal and ventral pathway activities in the social human brain. *cortex*, 58, 289-300.

Minor points:

Authors should show not only brain activation maps but also bar graphs of brain activation for supporting reader's intuitive understanding.

Review form: Reviewer 2

Recommendation

Accept with minor revision (please list in comments)

Scientific importance: Is the manuscript an original and important contribution to its field?

Excellent

General interest: Is the paper of sufficient general interest?

Good

Quality of the paper: Is the overall quality of the paper suitable?

Excellent

Is the length of the paper justified?

Yes

Should the paper be seen by a specialist statistical reviewer?

Yes

Do you have any concerns about statistical analyses in this paper? If so, please specify them explicitly in your report.

No

It is a condition of publication that authors make their supporting data, code and materials available - either as supplementary material or hosted in an external repository. Please rate, if applicable, the supporting data on the following criteria.

Is it accessible?

No

Is it clear?

No

Is it adequate?

Yes

Do you have any ethical concerns with this paper?

No

Comments to the Author

The manuscript "Re-imagining the intentional stance" by Abu-Akel and colleagues reports a very interesting and timely study which aimed at identifying neural correlates of adoption of the intentional stance. The experiment was inspired by earlier studies of Gallagher et al. (2002) and Chaminade et al. (2012) addressing similar questions. However, the present study aimed at de-confounding the way the intentional stance concept was operationalized in previous research. This was done in by orthogonal manipulation of active agency and humanness. In a 2x2 design, participants were led to believe that they were playing either with:

- an active competitive human,
- a human who followed passively a pre-defined script,
- an AI system capable of analyzing participants' moves
- or a passive computer program following a response script.

It was assumed that manipulation of the active agency addresses the concept of the intentional stance, while addressing the aspect of humanness is related to only interaction with natural vs. artificial agents.

The results showed that indeed broad mentalizing network (bilateral TPJ, precuneus, anterior PCC, right temporal lobe) was involved in the contrast of active vs. passive agents. The contrast of human vs. computer opponent yielded differential effects only in rTPJ and anterior PCC. In addition, a localizer ToM task was conducted and showed a large overlap with the activity related to the contrast of active vs. passive agency.

The authors conclude that their results show (in a non-confounded manner) that adopting intentional stance (regardless of whether it is a natural or artificial agent) involves a broad mentalizing network, while interacting with a natural (human) agent involves spontaneous default process of mentalizing, with activation of only a subset of the mentalizing network, namely rTPJ and anterior PCC.

Overall, I find the study very well-conceived, clear and elegant. The manuscript is well-written and it is a pleasurable read. The questions are timely and scientifically interesting. The results are clearly presented.

I only have concerns related to the general conceptualization of the study, and thereby, conclusions drawn from the results.

I am not entirely sure if contrasting a passive script-following agent with an actively-deciding agent is indeed a proper operationalization of Dennett's intentional stance concept. To me, this comparison addresses the issue of agency per se (not necessarily intentionality).

One can imagine a case in which a human is passively following a script because s/he is forced to do so. As an external observer, we would still attribute intentionality to him/her, maybe even inferring mental states that are in opposition to the observed behavior (s/he is desiring to perform an action that is against what s/he is told to do, but she is complying with the order because it's the most rational approach, etc.). Therefore a human agent that is passively following a script vs. a human agent acting in accordance to their own desires/will differ in the degree of agency, but not in intentionality.

Similarly, a sophisticated AI system which autonomously makes decision might not have any mental states attributed (thereby, intentional stance not adopted).

In this context, the activity observed in the first contrast of active vs. passive agents (independent of whether it was a human or a computer) – involving the broad mentalizing network – might not be related to the intentional stance but rather to attributed agency.

A second, more minor, point is related to the interaction effect in the left frontal pole. I was wondering if this effect might be due to the task itself. If this region has been linked to inhibitory control, or suppression of distractions, then perhaps the effect observed here is related to executive functions involved in the rock-paper-scissors game itself, for example, in the attempts of inhibiting a natural tendency to repeat the same move as the opponent. This kind of processes might be specifically strongly activated for human opponents in the active mode, as this is the

condition where participants might have been particularly competitive.

Decision letter (RSPB-2019-1553.R0)

12-Aug-2019

Dear Dr Abu-Akel:

I am writing to inform you that your manuscript RSPB-2019-1553 entitled "Re-imaging the intentional stance" has, in its current form, been rejected for publication in Proceedings B. This action has been taken on the advice of referees, who have recommended that substantial revisions are necessary. With this in mind we would be happy to consider a resubmission, provided the comments of the referees are fully addressed. However please note that this is not a provisional acceptance, and that your manuscript will be sent back out for review.

In particular, as you will see below, both reviewers applaud your goals, but question whether you have really disentangled intentionality (or whether it's something more like agency). Each raise important issues below that must be addressed prior to further consideration of your manuscript.

Sincerely,

Dr Sarah Brosnan
Editor, Proceedings B
mailto: proceedingsb@royalsociety.org

Associate Editor
Board Member: 1
Comments to Author:

Both reviewers are positive about this paper and see novelty in the findings. Reviewer 2 has some comments about interpretation that need to be addressed, specifically in relation to the concepts

of intentionality and agency. Reviewer 1 has some more serious concerns about the analysis that need to be addressed, either by justifying the current approach or making the suggested changes. The paper is well written and largely accessible to a broad audience, but the details the reviewers are asking for are important in determining whether the contribution is genuinely novel or largely incremental.

Reviewer(s)' Comments to Author:

Referee: 1

Comments to the Author(s)

The scope of this paper is interesting. However, if authors would like to separate two compound factors involved in "intentional stance", they should define these factors more strictly.

To my understanding, we must ideally prepared orthogonal independent variables in fMRI analysis (if not, these variables might eat same brain signals and the result might be distorted). Although, authors claimed they prepare two orthogonal independent variables, they should guarantee these two different variables are rigid and independent ones from experimental data.

Recent studies have suggested that strategic aspects of participant's behaviors also affected brain activities in interactive games. If authors purely want to argue about "the stance" of participants, they should conduct similar behavioral analysis in following previous articles and exclude brain signals caused by the behavioral difference among conditions. Even if authors claim that "stance" also contain behavioral aspects, they should do computational behavioral analysis in line with modern neuroscience.

[ref]

Hampton, A. N., Bossaerts, P., & O'Doherty, J. P. (2008). Neural correlates of mentalizing-related computations during strategic interactions in humans. *Proceedings of the National Academy of Sciences*, 105(18), 6741-6746.

Lee, D., McGreevy, B. P., & Barraclough, D. J. (2005). Learning and decision making in monkeys during a rock-paper-scissors game. *Cognitive Brain Research*, 25(2), 416-430.

Takahashi, H., Izuma, K., Matsumoto, M., Matsumoto, K., & Omori, T. (2015). The anterior insula tracks behavioral entropy during an interpersonal competitive game. *PloS one*, 10(6), e0123329.

Furthermore, there are few information about participants. The attitude toward "human" and "computer" were supposed to be varied among people. The various factors (e.g. education, personalities, culture...) shape our attitude and there are huge diversity in it. Hence if authors merely merge data regardless of individual differences, it is not so strange that the identities about opponent did not predict brain signal well (because S/N of data is too bad). Please describe precise information about participant's identities (e.g. education, Expertise) and authors should show the attitudes toward "human" and "computer" was not so varied in this experiment.

Did participants ask some questionnaires about the impression of opponents? Previous studies have suggested the impression of other agent lineally predicted the activation of mentalizing network. The resolutions of words "human" and "computer" are too low (e.g. the image of computer might to be different between participants with the knowledge about AlfaGO and people who did not know it). Authors should quantify participant's stance toward different two opponents.

{ref}

Takahashi, H., Terada, K., Morita, T., Suzuki, S., Haji, T., Kozima, H., ... & Naito, E. (2014). Different impressions of other agents obtained through social interaction uniquely modulate dorsal and ventral pathway activities in the social human brain. *cortex*, 58, 289-300.

Minor points:

Authors should show not only brain activation maps but also bar graphs of brain activation for supporting reader's intuitive understanding.

Referee: 2

Comments to the Author(s)

The manuscript "Re-imagining the intentional stance" by Abu-Akel and colleagues reports a very interesting and timely study which aimed at identifying neural correlates of adoption of the intentional stance. The experiment was inspired by earlier studies of Gallagher et al. (2002) and Chaminade et al. (2012) addressing similar questions. However, the present study aimed at de-confounding the way the intentional stance concept was operationalized in previous research. This was done in by orthogonal manipulation of active agency and humanness. In a 2x2 design, participants were led to believe that they were playing either with:

- an active competitive human,
- a human who followed passively a pre-defined script,
- an AI system capable of analyzing participants' moves
- or a passive computer program following a response script.

It was assumed that manipulation of the active agency addresses the concept of the intentional stance, while addressing the aspect of humanness is related to only interaction with natural vs. artificial agents.

The results showed that indeed broad mentalizing network (bilateral TPJ, precuneus, anterior PCC, right temporal lobe) was involved in the contrast of active vs. passive agents. The contrast of human vs. computer opponent yielded differential effects only in rTPJ and anterior PCC. In addition, a localizer ToM task was conducted and showed a large overlap with the activity related to the contrast of active vs. passive agency.

The authors conclude that their results show (in a non-confounded manner) that adopting intentional stance (regardless of whether it is a natural or artificial agent) involves a broad mentalizing network, while interacting with a natural (human) agent involves spontaneous default process of mentalizing, with activation of only a subset of the mentalizing network, namely rTPJ and anterior PCC.

Overall, I find the study very well-conceived, clear and elegant. The manuscript is well-written and it is a pleasurable read. The questions are timely and scientifically interesting. The results are clearly presented.

I only have concerns related to the general conceptualization of the study, and thereby, conclusions drawn from the results.

I am not entirely sure if contrasting a passive script-following agent with an actively-deciding agent is indeed a proper operationalization of Dennett's intentional stance concept. To me, this comparison addresses the issue of agency per se (not necessarily intentionality).

One can imagine a case in which a human is passively following a script because s/he is forced to do so. As an external observer, we would still attribute intentionality to him/her, maybe even inferring mental states that are in opposition to the observed behavior (s/he is desiring to perform an action that is against what s/he is told to do, but she is complying with the order because it's the most rational approach, etc.). Therefore a human agent that is passively following a script vs. a human agent acting in accordance to their own desires/will differ in the degree of agency, but not in intentionality.

Similarly, a sophisticated AI system which autonomously makes decision might not have any mental states attributed (thereby, intentional stance not adopted).

In this context, the activity observed in the first contrast of active vs. passive agents (independent of whether it was a human or a computer) – involving the broad mentalizing network – might not be related to the intentional stance but rather to attributed agency.

A second, more minor, point is related to the interaction effect in the left frontal pole. I was wondering if this effect might be due to the task itself. If this region has been linked to inhibitory control, or suppression of distractions, then perhaps the effect observed here is related to executive functions involved in the rock-paper-scissors game itself, for example, in the attempts of inhibiting a natural tendency to repeat the same move as the opponent. This kind of processes might be specifically strongly activated for human opponents in the active mode, as this is the condition where participants might have been particularly competitive.

Author's Response to Decision Letter for (RSPB-2019-1553.R0)

See Appendix A.

RSPB-2020-0244.R0

Review form: Reviewer 1

Recommendation

Accept as is

Scientific importance: Is the manuscript an original and important contribution to its field?

Good

General interest: Is the paper of sufficient general interest?

Good

Quality of the paper: Is the overall quality of the paper suitable?

Good

Is the length of the paper justified?

Yes

Should the paper be seen by a specialist statistical reviewer?

No

Do you have any concerns about statistical analyses in this paper? If so, please specify them explicitly in your report.

No

It is a condition of publication that authors make their supporting data, code and materials available - either as supplementary material or hosted in an external repository. Please rate, if applicable, the supporting data on the following criteria.

Is it accessible?

Yes

Is it clear?

Yes

Is it adequate?

Yes

Do you have any ethical concerns with this paper?

No

Comments to the Author

After this revision, I have judged that this paper was suitable for the publication.

Decision letter (RSPB-2020-0244.R0)

18-Mar-2020

Dear Dr Abu-Akel

I am pleased to inform you that your manuscript RSPB-2020-0244 entitled "Re-imaging the intentional stance" has been accepted for publication in Proceedings B.

If you have not already done so, please submit the following:

1) A media summary: a short non-technical summary (up to 100 words) of the key findings/importance of your manuscript.

2) Data accessibility section and data citation

[http://datadryad.org/submit?journalID=RSPB&manu=\(Document not available\)](http://datadryad.org/submit?journalID=RSPB&manu=(Document not available)) which will take you to your unique entry in the Dryad repository. If you have already submitted your data to dryad you can make any necessary revisions to your dataset by following the above link.

Please see <https://royalsociety.org/journals/ethics-policies/data-sharing-mining/> for more details.

For more information on our Licence to Publish, Open Access, Cover images and Media summaries, please visit <https://royalsociety.org/journals/authors/author-guidelines/>.

Once again, thank you for submitting your manuscript to Proceedings B and congratulations on your acceptance. If you have any questions at all, please do not hesitate to get in touch.

Sincerely,

Dr Sarah Brosnan
Editor, Proceedings B
mailto: proceedingsb@royalsociety.org

Associate Editor

Board Member

Comments to Author:

In my view this paper makes a valuable contribution to our understanding of intentionality and theory of mind and the utility of methods assessing these capacities.

Reviewer(s)' Comments to Author:

Referee: 1

Comments to the Author(s).

After this revision, I have judged that this paper was suitable for the publication.

Decision letter (RSPB-2020-0244.R1)

20-Mar-2020

Dear Dr Abu-Akel

I am pleased to inform you that your manuscript entitled "Re-imaging the intentional stance" has been accepted for publication in Proceedings B.

Open Access

Paper charges

Sincerely,

Editor, Proceedings B
mailto: proceedingsb@royalsociety.org

Appendix A

Feb 4, 2020

Prof. Sarah Brosnan
Editor
Royal Society: Proceedings B

Dear Prof. Brosnan,

Thank you for considering our manuscript (ms# RSPB-2019-1553) titled "Re-imaging the intentional stance". We thank you and the referees for the thoughtful comments and suggestions. We read these comments carefully and revised the manuscript accordingly. Please find below our point-by-point response to each comment. We hope that you will find the revised manuscript acceptable for publication in *Proceedings B*.

Sincerely,
Ahmad Abu-Akel

Response to reviews

Associate Editor; Board Member: 1

1. Both reviewers are positive about this paper and see novelty in the findings. Reviewer 2 has some comments about interpretation that need to be addressed, specifically in relation to the concepts of intentionality and agency. Reviewer 1 has some more serious concerns about the analysis that need to be addressed, either by justifying the current approach or making the suggested changes. The paper is well written and largely accessible to a broad audience, but the details the reviewers are asking for are important in determining whether the contribution is genuinely novel or largely incremental.

1R. We thank you and the reviewers for the positive assessment our paper. We have carefully considered all comments and revised the manuscript accordingly. We hope this has helped clarify why this is a substantial and novel contribution, separating out factors that have previously been confounded in previous studies, and so advancing understanding in this influential approach to theorizing and studying social cognition in the brain.

Referee: 1

2. The scope of this paper is interesting. However, if authors would like to separate two compound factors involved in "intentional stance", they should define these factors more strictly. To my understanding, we must ideally prepared orthogonal independent variables in fMRI analysis (if not, these variables might eat same brain signals and the result might be distorted). Although, authors claimed they prepare two orthogonal independent variables, they should guarantee these two different variables are rigid and independent ones from

experimental data.

2R. We thank the reviewer for finding our paper interesting. We apologize for any lack of clarity here regarding the concept and operationalization of the factors involved in this experiment. We designed this study such that the manipulations that are overtly presented, namely (a) whether the participant believes they are playing against a human opponent or a computer opponent (the factor we have called “embodiment” in the manuscript) and (b) whether the participant believes the opponent to be acting with free agency (“active”) or no agency (“passive”) (the factor we have called “intentionality” in the manuscript) to be **notionally** independent of each other. To be clear here, there are two experimental factors. These two factors are **notionally** independent, or orthogonal, to each other. They are well defined in this regard and clearly experimentally independent. This is therefore a standard 2x2 factorial design of experiment no different to many thousands of others undertaken and published. However, what the reviewer notes in respect of potential brain response is of course quite true. The fact that the input manipulation is notionally orthogonal does not by any means guarantee that the detected fMRI BOLD response will also be orthogonal. At one extreme, perhaps the brain is insensitive to the active/passive difference. But that is exactly the point here. We perform the experiment with notionally orthogonal input manipulations precisely in order to see if these manipulations are reflected in the fMRI signal, or whether there is an interaction between these two factors. And indeed we do seem to find an interaction (in the left frontal pole) indicating that even though the experimental factors are notionally independent (orthogonal) the brain does not see them as being totally independent. The reviewer is also correct that, in the worst case, if the two factors were totally correlated in the brain then the fMRI signal would be identical between the factors and we would have no ability to distinguish between them. But again, we would argue that this is exactly the point of the experiment. If we were to find this case (which we did not) then it would have shown that the notionally independent experimental factors were not at all independent. That would be interesting new information itself worthy of publication. In summary, we stand by the design of the experiment and would argue that is concordant with multiple other published fMRI studies.

The confusion here perhaps lies with the terminology used in the manuscript. We have therefore modified the manuscript by the addition of the following sentence on page 6:

“It should be noted of course that orthogonality of the experimental factors does not necessarily imply that the measured fMRI BOLD signal responses will themselves be orthogonal.”

3. Recent studies have suggested that strategic aspects of participant's behaviors also affected brain activities in interactive games. If authors purely want to argue about "the stance" of participants, they should conduct similar behavioral analysis in following previous articles and exclude brain signals caused by the behavioral difference among conditions. Even if authors claim that "stance" also contain behavioral aspects, they should do computational behavioral analysis in line with modern neuroscience.

3R. We thank the reviewer for the interesting comments and suggestions here. Firstly, we would argue that this paper is principally a standalone fMRI experiment that seeks to disambiguate brain responses to the notionally independent factors of “embodiment” and “intentionality” as discussed above. The paper has a single focus that is interesting and of merit by itself. Secondly, naturally having found significant results it would of course be of interest in future to conduct further follow up studies – behavioural, computational and further fMRI – to better understand the underlying processes involved. We are completely in agreement with the reviewer here. However we would argue this falls outside the scope of this paper and is not necessary in order to support the (limited) findings that we advance in the manuscript.

Finally, I think there is a danger of being overly focused on the semantics of “the stance”. This is not a neurobiological term. So, I think we would agree with the reviewer that we **don’t** want to purely argue about “the stance”. We have an incorporated use of it from Dennet as it is in current parlance and because it provides a base theoretical background to this study. We note however that there is at least one confound within the concept of “intentional stance” - the possible distinction between intentionality and embodiment (page 3). The very purpose of this paper was to explore this potential confound. But we make no overarching claim to understand “the stance” or to provide a fuller neurobiological understanding here.

4. Furthermore, there are few information about participants. The attitude toward "human" and "computer" were supposed to be varied among people. The various factors (e.g. education, personalities, culture...) shape our attitude and there are huge diversity in it. Hence if authors merely merge data regardless of individual differences, it is not so strange that the identities about opponent did not predict brain signal well (because S/N of data is too bad). Please describe precise information about participant's identities (e.g. education, Expertise) and authors should show the attitudes toward "human" and "computer" was not so varied in this experiment.

4R. We thank the reviewer for their comments here.

Firstly, we would draw the reviewer’s attention to the statement about participant information on page 7:

“2.1. Participants: 24 right-handed, English-proficient healthy adults (5 males; 19 females; mean age (SD) = 21.21±4.21) participated in the study. All participants were students from the University of Birmingham. Exclusion criteria included having a history of psychiatric illness, epilepsy, neurological disorders, brain injury as well as current alcohol or substance abuse problems.”

(Note the sentence in red has been added for clarity).

Such selection criteria are fairly typical for characterizing putatively normative behavior and are a common accepted standard in the majority of published fMRI work. However, as with any experiment, the limitations of generalizing the group average results too far beyond the testing

group should be apparent.

The reviewer is of course correct in realizing that there will be individual differences between participants and that these will likely be reflected both in their individual behavior and measured fMRI response. However, this is true of all fMRI group studies. Our experiment and analysis is entirely concordant with modern fMRI experimental practice and data analysis. It is important to realize the implication of the particular highest level fMRI analysis conducted (section 2.6 of manuscript). A 2x2 repeated measures ANOVA analysis was performed. The significance of this design is that it deliberately accounts for individual differences, greatly increasing the statistical power, and only shows group average results for the experimental factors of interest (intentionality, embodiment, and the interaction between the two) if they exist. (Much like a one group paired t-test is far more powerful than a two group unpaired t-test). In this design of experiment, particular details of any one individual (e.g. age, sex, handedness, educational level etc) are not of importance as they are effectively subtracted out in the within-subject design. Here we are interested only in the group average response to the experimental factors. To that extent, this is a deliberately limited and simple analysis. We therefore stand by the design of experiment and analysis conducted as being appropriate and sufficient for purpose and, as mentioned, entirely consistent with established practice. To demonstrate group average effects by factor it is not necessary to list all aspects of the individuals participating.

It is also of course possible, very likely even, that some aspects of the individual will be predictive for their behavior and neural response, even simple things like age or sex. There is considerable merit in pursuing such interesting questions in future. But such studies would be to examine the role of individual differences in ToM processing and not the group average effects. This is outside the scope of this paper and was not the question of interest for us here.

5. Did participants ask some questionnaires about the impression of opponents? Previous studies have suggested the impression of other agent lineally predicted the activation of mentalizing network. The resolutions of words "human" and "computer" are too low (e.g. the image of computer might to be different between participants with the knowledge about AlfaGO and people who did not know it). Authors should quantify participant's stance toward different two opponents.

{ref}
Takahashi, H., Terada, K., Morita, T., Suzuki, S., Haji, T., Kozima, H., ... & Naito, E. (2014). Different impressions of other agents obtained through social interaction uniquely modulate dorsal and ventral pathway activities in the social human brain. *cortex*, 58, 289-300.

R5. The reviewer raises some interesting questions. Firstly, as mentioned on page 9 of the manuscript we did conduct a structured post-scanning briefing during which participants were asked how they understood and experienced the experimental conditions. We used this purely to confirm that the participants believed (or at least said they believed) in the identity (that they truly were playing against a human or a computer) and intentionality (that the role was

active or passive) of the experimental conditions. None of the participants expressed doubt about the perceived reality of the condition. Had a participant expressed doubt we would have excluded them from further analysis. However no finer-grained information was collected. At the time it was considered that the participants would be binarized in their perception of the experiment; they either believed the manipulations they were presented with, or they were suspicious of the whole procedure. Fortunately, no-one was in the second group.

In respect of the comment about previous studies predicting the degree of activation of the mentalizing network: this is an interesting paper and we have now cited it in our introduction (see page 4), and the discussion (see page 16).

In respect of the comment regarding the resolution between “human” and “computer”: on a practical point, the particular condition was indicated to the participant by means of several intro screens that were displayed prior to each block beginning. The first was a “getting ready screen”. For example this would say:

“HUMAN OPPONENT GETTING READY...

PLEASE WAIT”

and would be on screen for a variable amount of time to reinforce the perception of a real human opponent being prepared. This would then be followed by a further instruction screen, for example:

“HUMAN OPPONENT - ACTIVE

In this block you will play ten rounds against the researcher. He will be trying his best to outwit you using all his skills. After the countdown you will both privately select an item. Once you have both made a choice, they will be revealed at the same time.

If you win a round, you will be awarded 1 point."

Both instruction screens were presented in a large font and were easy to read. No participant commented on a problem with visually identifying the condition. I am conscious that the reviewer may have meant here that the resolution distinction was not a visual one (could they see the instructions) but rather a conceptual one (to what extent were the conditions separated in the mental world of the participant). This is perhaps a deeper question. We would note that one way of treating this is yet again as an individual difference. Yes, possibly for some participants the mental distinction between playing a computer vs playing a human is huge but for others the difference was not so big. And the same argument can be used with the condition difference of active vs passive. But these are all examples of individual differences. As explained above, the specific analyses we conducted at highest level using a repeated measures 2x2 ANOVA design allows us to account for and remove all such individual differences. We are then left with the results that we did in fact find. Figure 2 and Table 1 shows the main effect of

the RPS task. For example merely instructing the participants that they were playing a human opponent (as opposed to playing a computer program) causes **on average** a hyperactivation of the right TPJ. We would argue this is of interest and significant.

On the specific point regarding differences between participants with the knowledge about AlfaGO (sic AIRPS?) and people who did not know it. No-one knew about AIRPS as the entire concept and name was fictional and made up for the purpose the experiment. All participants were therefore in equal state of knowledge here. We have now amended the manuscript on page 8 to make clear AIRPS was a fictional construct.

6. Minor points: Authors should show not only brain activation maps but also bar graphs of brain activation for supporting reader's intuitive understanding.

R6. The reviewer makes a good suggestion. However, to implement this as suggested would require identifying specific regions of interest (ROI) for each activated area for the main analyses. This would require “double-dipping” or re-use of our data: firstly in the group analysis to show peaks and create ROIs for each peak, and then a re-use of these ROIs to examine the behavior within. This is considered bad practice within the fMRI community and would normally be criticized by reviewers. It is however acceptable to use an independent task such as the ToM localizer to create the ROIs and then examine the behavior of the main task within these. However the ToM localizer results only partially overlap with the main experiment so this would not fulfill the request from the reviewer. We therefore do not include bar graphs.

Referee: 2

7. The manuscript “Re-imagining the intentional stance” by Abu-Akel and colleagues reports a very interesting and timely study which aimed at identifying neural correlates of adoption of the intentional stance. The experiment was inspired by earlier studies of Gallagher et al. (2002) and Chaminade et al. (2012) addressing similar questions. However, the present study aimed at de-confounding the way the intentional stance concept was operationalized in previous research. This was done in by orthogonal manipulation of active agency and humanness. In a 2x2 design, participants were led to believe that they were playing either with:

- an active competitive human,
- a human who followed passively a pre-defined script,
- an AI system capable of analyzing participants’ moves
- or a passive computer program following a response script.

It was assumed that manipulation of the active agency addresses the concept of the intentional stance, while addressing the aspect of humanness is related to only interaction with natural vs. artificial agents.

The results showed that indeed broad mentalizing network (bilateral TPJ, precuneus, anterior PCC, right temporal lobe) was involved in the contrast of active vs. passive agents. The contrast of human vs. computer opponent yielded differential effects only in rTPJ and anterior PCC. In

addition, a localizer ToM task was conducted and showed a large overlap with the activity related to the contrast of active vs. passive agency.

The authors conclude that their results show (in a non-confounded manner) that adopting intentional stance (regardless of whether it is a natural or artificial agent) involves a broad mentalizing network, while interacting with a natural (human) agent involves spontaneous default process of mentalizing, with activation of only a subset of the mentalizing network, namely rTPJ and anterior PCC.

Overall, I find the study very well-conceived, clear and elegant. The manuscript is well-written and it is a pleasurable read. The questions are timely and scientifically interesting. The results are clearly presented.

7R. We thank the reviewer for the positive feedback and assessment of our manuscript.

8. I only have concerns related to the general conceptualization of the study, and thereby, conclusions drawn from the results. I am not entirely sure if contrasting a passive script-following agent with an actively-deciding agent is indeed a proper operationalization of Dennett's intentional stance concept. To me, this comparison addresses the issue of agency per se (not necessarily intentionality). One can imagine a case in which a human is passively following a script because s/he is forced to do so. As an external observer, we would still attribute intentionality to him/her, maybe even inferring mental states that are in opposition to the observed behavior (s/he is desiring to perform an action that is against what s/he is told to do, but she is complying with the order because it's the most rational approach, etc.). Therefore a human agent that is passively following a script vs. a human agent acting in accordance to their own desires/will differ in the degree of agency, but not in intentionality.

Similarly, a sophisticated AI system, which autonomously makes decision might not have any mental states attributed (thereby, intentional stance not adopted).

In this context, the activity observed in the first contrast of active vs. passive agents (independent of whether it was a human or a computer) – involving the broad mentalizing network – might not be related to the intentional stance but rather to attributed agency.

8R. We appreciate the reviewer's thoughtful and interesting comment but we do not agree that it raises any fundamental problem for our study or its interpretation.

The reviewer is correct that one can imagine a case in which a human is forced to follow a script, which invites rich inferences about their inner thoughts and feelings about being placed in this predicament. But this was not the case in our study because there was no suggestion that the human target was placed in such a predicament – i.e., there was no suggestion of coercion, or anything else that implied that s/he lacked basic agency to act/not act. What s/he lacked was the possibility of acting strategically within the context of the task, and so s/he was not a legitimate target for the intentional stance with respect to the task. We agree with the reviewer that participants may, for other reasons, have thought about the mental states of the human targets, but no more so in the human-passive condition than in the human-active condition. Therefore we think there are strong grounds to suppose that participants reasoned

more about intentional mental states of the target the human-active than in the human-passive condition, and if there were also differences in agency, they were just those differences that were relevant to the target being free to devise their own strategies for playing the game, which is the essence of their being a valid target for adopting the intentional stance (see quote from Dennet on p5 of the manuscript).

With respect to the AI system, we agree with the reviewer that participants may indeed have viewed a sophisticated AI system as having agency, and this was certainly our intention, because as Dennett's definition of the intentional stance makes clear, a target's agency is a prerequisite for application of the intentional stance. But we do not agree that any problems follow from this. The reviewer suggests that the AI system may not REALLY have mental states, just agency. The point of Dennett's theory is to render this kind of reasoning moot. According to Dennett, ALL THERE IS to being an intentional system is to be a system whose behavior is well-explained by adopting the intentional stance. There are no further questions to be asked about whether there really are mental states in the target [see also Thellman, S., Silvervarg, A., and Ziemke, T. (2017). Folk-psychological interpretation of human vs. humanoid robot behavior: exploring the intentional stance toward robots. *Front. Psychol.* 8:1962. doi: 10.3389/fpsyg.2017.01962; Marchesi S, Ghiglino D, Ciardo F, Baykara E, Wykowska A. Do we adopt the Intentional Stance towards humanoid robots?. *Frontiers in psychology.* 2019;10:450.]. We highlight this point in the introduction (page 5). Of course this view is debated among philosophers, but it is the essence of the intentional stance theory that we are testing in the present study, and so we think our design captures the intended contrasts adequately.

In sum, our objective was to resolve a critical confound in existing studies by orthogonalizing whether the target was natural-artificial and the degree to which the target was a legitimate target for the intentional stance. As the reviewer notes, the concepts of intentionality and agency are closely related, and the intentional stance theory itself is debated. However, it is hugely influential, both in theory and in experimental cognitive neuroscience, and we believe our study's design achieves its intended purposes.

We have not addressed this point any further in the current revisions because we are concerned about appearing to raise a straw man argument, which may confuse more readers than it helps, but we are very happy to be guided by the reviewer and editor in this decision.

9. A second, more minor, point is related to the interaction effect in the left frontal pole. I was wondering if this effect might be due to the task itself. If this region has been linked to inhibitory control, or suppression of distractions, then perhaps the effect observed here is related to executive functions involved in the rock-paper-scissors game itself, for example, in the attempts of inhibiting a natural tendency to repeat the same move as the opponent. This kind of processes might be specifically strongly activated for human opponents in the active mode, as this is the condition where participants might have been particularly competitive.

9R. We thank the reviewer for introducing a potential explanation for the interaction effect we observed in the left frontal pole. We added this interpretation in the discussion on page 19:

“One can also conjecture that the observed effect for the frontal pole is due to executive functions involved in the rock-paper-scissors game itself, for example, in the attempts of inhibiting a natural tendency to repeat the same move as the opponent, which would be consistent with the role of the frontal pole in manipulating and maintaining information from self-generated behavior (Christoff & Gabrieli, 2000). Such processes might be specifically strongly activated where participants might have been particularly competitive.”